# Quantitative Gait Analysis of Patients with Severe Symptomatic Spinal Stenosis Utilizing the Gait Profile Score: An Observational Clinical Study

**DOI:** 10.3390/s22041633

**Published:** 2022-02-19

**Authors:** Jan Lodin, Marek Jelínek, Martin Sameš, Petr Vachata

**Affiliations:** 1Neurosurgical Department, J. E. Purkyně University, Masaryk Hospital of Krajská Zdravotní a.s., Sociální Péče 3316/12A, 400 11 Ústí nad Labem, Czech Republic; martin.sames@kzcr.eu (M.S.); vachata@gmail.com (P.V.); 2Faculty of Medicine in Plzeň, Charles University, Husova 3, 306 05 Plzeň, Czech Republic; 3Laboratory for the Study of Movement, Faculty of Health Studies, J. E. Purkyně University in Ústí nad Labem, Pasteurova 3544/1, 400 96 Ústí nad Labem, Czech Republic; mrkjelda@seznam.cz

**Keywords:** lumbar spine stenosis, kinematic parameters, gait analysis, spatiotemporal parameters, claudication, optoelectronic systems

## Abstract

Lumbar spine stenosis (LSS) typically manifests with neurogenic claudication, altering patients’ gait. The use of optoelectronic systems has allowed clinicians to perform 3D quantitative gait analysis to quantify and understand these alterations. Although several authors have presented analysis of spatiotemporal gait parameters, data concerning kinematic parameters is lacking. Fifteen patients with LSS were matched with 15 healthy controls. Quantitative gait analysis utilizing optoelectronic techniques was performed for each pair of subjects in a specialized laboratory. Statistical comparison of patients and controls was performed to determine differences in spatiotemporal parameters and the Gait Profile Score (GPS). Statistically significant differences were found between patient and control groups for all spatiotemporal parameters. Patients had significantly different overall GPS (*p* = 0.004) and had limited internal/external pelvic rotation (*p* < 0.001) and cranial/caudal movement (*p* = 0.034), limited hip extension (*p* = 0.012) and abduction/adduction (*p* = 0.012) and limited ankle plantar flexion (*p* < 0.001). In conclusion, patients with LSS have significantly altered gait patterns in three regions (pelvis, hip and ankle) compared to healthy controls. Analysis of kinematic graphs has given insight into gait pathophysiology of patients with LSS and the use of GPS will allow us to quantify surgical results in the future.

## 1. Introduction

Lumbar spine stenosis (LSS) is a common degenerative condition predominantly affecting patients over the age of 65 [1]. A hallmark of LSS is the presence of neurogenic claudication, defined as the paresthesia, dysesthesia, radiating from the buttocks distally [2]. Clinically, this manifests as gait dysfunction, which results in the decline of patient quality of life. Quantifying gait disability remains a key goal for clinicians treating patients with LSS, as it would allow them to assess the severity of the disease, monitor disease progression and analyze surgical success. As such, many have utilized modern motion capture and optoelectronic systems to perform complex 3D gait reconstructions, allowing detailed analysis of separate anatomical areas during the gait cycle. However, the complexity of 3D movement patterns limits their interpretation and implementation into clinical practice. Although several papers have published the results of gait analyses of patients with LSS, most are limited to analyzing spatiotemporal gait parameters [3]. These parameters are descriptive of the gait cycle and its characteristics, such as walking speed, cadence, step length, etc. As such, they give us limited information on changes within individual motion segments, for example how lumbar stenosis affects motion in major joints of the lower limbs. Thus, a small number of studies analyzed kinematic angular parameters, which describe angle changes within a specific joint in a set of anatomical planes [4,5]. Unfortunately, none of these papers have provided quantitative assessment of kinematic parameters using a validated gait scoring system, which could allow objective inter-patient comparison. The present paper is a single-institution, observational, cross-sectional study made up of 15 patients with severe spinal canal stenosis of the lower lumbar spine, who underwent 3D quantitative motion analysis prior to surgical decompression. To the best of our knowledge, it is the first study to describe 3D kinematic data of patients with LSS using the validated Gait Profile Score (GPS). This allowed us to quantify and depict significant gait alterations in three distinct regions (pelvis, hip, and ankle), upon which clinicians can focus on when performing gait analysis of patients with LSS.

## 2. Materials and Methods

### 2.1. Aims and Hypothesis

Our hypothesis was that gait patterns of patients with severe symptomatic spinal stenosis are significantly altered compared to healthy controls. Our aim was to analyze specific differences in spatiotemporal and kinematic gait parameters between patients with severe spinal stenosis and healthy controls. Furthermore, our goal was to depict these differences using a validated gait scoring system in order to objectively quantify these differences. As such, we utilized the Gait Profile Score (GPS), which summarizes gait kinematics in the form of a whole number and can be compared between subjects. Our final aim was to identify gait alterations in specific motion segments using Gait Variable Scores (GVS), a subset of nine variables which form the GPS, and visualize these alterations using motion graphs.

### 2.2. Patient and Control Selection Criteria

Patients were selected based on a combination of clinical, radiological, and medical history criteria. Clinical criteria included symptoms of severe spinal stenosis, specifically the presence of typical neurogenic claudication (paresthesia, dysesthesia, weakness, numbness during gait), with a claudication interval below 500 m. Segmental lower limb paresis was an exclusion criterion, as it could potentially affect gait patterns. All patients fulfilled the Oswestry Disability Index questionnaire and a severe disability score (ODI > 40%) was required for them to be admitted into the study. Radiological criteria consisted of a lumbar spine MRI, which demonstrated the presence of absolute spinal canal stenosis of Schizas grade C or D [6]. Only patients with spinal stenosis in the regions L3-S1 were included. Furthermore, all patients had full-body upright and dynamic X-rays performed, to rule out the presence of spine deformities, spondylolisthesis or latent instability of the stenotic lumbar segments. Furthermore, hip X-rays were performed to rule out coxarthrosis. Finally, a detailed patient history was obtained to identify possible exclusion criteria. These included previous spine, hip or knee trauma/surgery, ischemic lower limb disease, psychiatric or neurodegenerative disorders. All patients signed informed consent to undergo 3D motion analysis and allowing future publishing of their anonymized data.

Controls were selected from local hospital and university staff and were matched by gender, age, and body mass index with individual patients. All controls were required to be completely healthy, without the presence of chronic diseases or previous trauma/surgery in their medical history.

### 2.3. 3D Quantitative Motion Analysis

All patients underwent 3D quantitative motion analysis in a specialized laboratory, equipped with 11 infrared cameras Oqus 300 and 300+, placed in equal intervals along the laboratory walls. Furthermore, two force platforms (Kistler type 9281EA, Kistler Group, Winterthur, Switzerland) 400 × 600 × 100 mm were placed on the laboratory floor (Figure 1).

The subjects were then marked with passive 19 mm reflective markers, which were placed individually or in clusters of four above major anatomical structures (Table 1), via adhesive tape by an experienced anatomist, based on the atlas of Sint et al. [7] (Figure 2).

A total of 50 markers were used for static system calibration and 42 for dynamic analysis. After initial calibration in the standing position, the patient was asked to walk at a natural pace across the length of the laboratory. Kinematic data was then recorded via the Qualisys (Qualisys AB, Göteborg, Sweden) optoelectronic camera system at 100 Hz and then processed using QTM (Qualisys Track Manager) version 2020.3, PAF (Project Automation Framework) Gait Visual3D version 1.4.2.83 (C-Motion, Bethesda, MD, USA) and Visual3D x64 Professional version 6.03.6 software. A 6 Hz low-pass filter was used to reduce noise from marker positions. The CAST model derived by Cappozzo et al. was used to process movements of individual body segments [8]. The subjects underwent a total of 10 walks across the length of the laboratory, which resulted in 14–16 complete gait cycles being recorded. A minimum of 15 complete gait cycles was the minimal inclusion criterion. Unfortunately, plantar force data was not consistently obtained, due to decreased step length of patients with spinal stenosis. This led to suboptimal placement of the foot sole on force platforms, resulting in erroneous recordings.

### 2.4. Spatiotemporal and Kinematic Gait Parameters

Two sets of gait parameters were obtained for each patient and control: spatiotemporal and kinematic angular. Spatiotemporal parameters consisted of descriptive characteristics of the gait cycles, which were recorded in basic SI units as mean averages throughout 15 gait cycles (Table 2).

Kinematic angular parameters describe angle changes of a joint in one anatomical plane throughout the gait cycle. In order to objectively quantify these changes, we utilized the Gait Profile Score (GPS), an index first described by Baker et al., which presents kinematic changes globally in the form of a whole number [9]. This allowed comparison of several kinematic parameters simultaneously between two groups of subjects, as well as quantifying the extent of this difference. Furthermore, the GPS was broken down into nine Gait Variable Scores (GVS), which are calculated as a root mean square difference between a patient’s sagittal, axial, and frontal gait curve for each major lower limb joint and normative data representative of the population, which was initially incorporated within the Qualysis. As such, they represent angular changes within major joints in multiple anatomic planes, which were then used to create a Movement Analysis Profile (MAP) for each patient (Figure 3, Table 3).

GVS data was recorded in the form of motion graphs for each major joint in multiple planes throughout 15 gait cycles. The Qualisys software then fused these multiple consistency graphs into an average representative motion graph for each motion segment (Figure 4).

These average motion graphs were then used to generate individual GVSs, which in turn form the final GPS. Kinematic angular data of controls was then compared to normative kinematic angular data, representative of the mean population, which is initially incorporated within the Qualisys software. Control kinematic data was required to fall within one standard deviation of normative data, in order to be representative of the mean population. Six GVS components are side oriented, as they focus on major joints within both lower limbs. In these cases, mean right and left sided values, as well as combined values of both limbs were calculated throughout the gait cycle and the combined values were used for statistical analysis. In cases where statistically significant differences were found between GVS parameters of patients and controls, detailed analysis of motion graphs was performed to identify specific kinematic alterations within a joint throughout the gait cycle.

### 2.5. Statistical Methods

Statistical analysis was performed using the Statistika 13.0 (TINCO software Inc., Palo Alto, CA, USA) software. Comparison of mean spatiotemporal parameters, GPS and GVS values was performed between two individual groups- the patient group (PG) and their matched control group (CG). All patients were matched by sex, age and BMI to their respective controls. Initially, the Shapiro–Wilks test was used to test for normality. Afterwards, the *F*-test was used to test the null hypothesis that both the patient and control group had equal variance. Based on whether the null hypothesis was rejected or not, either the paired sample *t*-test or Welch’s *t*-test were used to determine statistically significant differences in spatiotemporal parameters, GPS and GVS values between the patient and control groups. Pearson and Spearman rank correlations were performed between individual gait parameters, ODI scores and age. Significance levels were set at α = 0.05 for statistical tests.

## 3. Results

### 3.1. General Information and Demographics

A total of 594 patients were admitted to our department for lumbar spine surgery due to degenerative lumbar spine disease, in the time period from 01.01.2019 to 31.12.2020. Of these patients, 202 had primary lumbar spine stenosis (LSS) of Schizas grade C or D. A total of 41 patients were eliminated due the presence of radicular pain or segmental L4, L5 or S1 paresis. A further 28 were eliminated due to a history of hip/knee surgery or arthrosis grade III/IV, 19 due to having had prior lumbar spine surgery, 12 due to having spinal stenosis above the L3 vertebra, 11 due to a history of cervical spondylotic myelopathy, 7 due to a history of ischemic lower limb disease and 9 did not give consent. Finally, 50 patients could not be considered for 3D quantitative motion analysis, as they underwent lumbar spine surgery during the COVID-19 pandemic with active government restrictions on unnecessary patient travel. The final cohort consisted of 15 patients (4 females and 11 males), mean age of 62.3, mean body mass index (BMI) 32.0, all of whom successfully underwent 3D quantitative motion analysis 2–4 weeks prior to lumbar decompression surgery. The most common stenotic level was L4/5 (14 cases), with five patients having isolated stenosis of L4/5, seven patients a combination of L3/4 and L4/5 stenosis and two patients having a combination of L4/5 and L5/S1 stenosis. One patient had isolated stenosis of L3/4. Twelve patients had spinal stenosis of Schizas grade D and three patients of Schizas grade C. Claudication intervals ranged from 10–500 m with radicular hypesthesia present in seven cases. Basic patient information is shown in Table 4.

### 3.2. Spatiotemporal Gait Parameters

An overview of measured spatiotemporal parameters with their definitions is shown in Table 2. Statistical analysis revealed statistically significant differences of all spatiotemporal parameters between patients and controls. Step length was significantly shorter in the patient group compared to the controls (PG = 0.50 m, CG = 0.68, *p* < 0.001) as was consequently stride length (*p* = 1.00 m, c = 1.35 m, *p* < 0.001). Conversely, stride width was significantly wider in the patient cohort compared to the controls (PG = 0.14 m, CG = 0.10 m, *p* = 0.004). Patients also demonstrated longer step times compared to healthy controls (PG = 0.62 s, CG = 0.54 s, *p* = 0.004), which resulted in a lower cadence (PG = 98.2 steps/min, CG = 111.2 steps/min, *p* = 0.003). Furthermore, compared to the control group, patients demonstrated significantly longer stance times (PG = 0.85 s, CG = 0.66 s, *p* < 0.001) and shorter swing times (PG = 0.40 s, CG = 0.42 s, *p* = 0.04) of the gait cycle. Finally, patients required longer durations of initial double limb support compared to healthy controls (PG = 0.22 s, CG = 0.12 s, *p* < 0.001). Pearson correlation coefficients demonstrated significant negative correlations (−0.51) between step and stride length with age (*p* = 0.48).

### 3.3. Kinematic Gait Parameters

An overview of measured kinematic parameters with their definitions is shown in Table 3. Statistically significant differences were found between the overall GPS between patients and controls (PG = 8.65, CG = 6.44, *p* = 0.004). The GPS was then broken down into individual GVSs, of which several showed statistically significant differences between the patient and control group. Hip flexion/extension significantly differed between patients and controls (PG = 6.22, CG = 11.31, *p* = 0.012) with kinematic graph analysis demonstrating restricted hip extension during the stance phase of the gait cycle (Figure 5).

Statistically significant differences were also found in hip abduction/adduction (PG = 5.73, CG = 3.73, *p =* 0.012), with kinematic graphs showing a decreased range of both abduction/adduction, more so for abduction, which was lower throughout the entire gait cycle (Figure 6).

Significant differences were also found between both pelvic cranial/caudal tilt (PG = 3.20, CG = 1.82, *p =* 0.034) and pelvic internal/external rotation (PG = 5.31, CG = 3.44, *p <* 0.001). Kinematic graphs demonstrated atypical ‘flat curves’, representing a lower range of pelvic motion for both pelvic parameters (Figure 7).

Finally, significant differences were found between ankle plantar/dorsal flexion (PG = 8.28, CG = 4.80, *p <* 0.001), with kinematic graphs showing a major decrease in maximal plantar flexion (Figure 8).

Kinematic parameters of the knee joints did not demonstrate significant differences between the patient and control group (PG = 8.46, CG = 9.74, *p =* 0.192) (Figure 9).

No statistically significant correlations were found between kinematic parameters with ODI scores or age.

## 4. Discussion

The main purpose of this paper was to objectively analyze differences in spatiotemporal and kinematic gait parameters between 15 patients with LSS and their matched controls. Significant differences were found between all spatiotemporal parameters as well as kinematic parameters in three regions (the pelvis, hip and ankle).

### 4.1. Gait Analysis Technique and Patient Selection

LSS is a common cause of movement disorders, especially in the elderly population [10]. Although clinicians currently utilize various imaging techniques to diagnose the presence and severity of LSS, their options of evaluating functional patient status are mostly limited to specialized questionnaires. Most commonly, they utilize the Oswestry Disability Index (ODI), which has been externally validated, but remains partially subjective. Although the ODI is an excellent tool for patient comparison and monitoring, it fails to analyze specific causes of functional impairment. Conversely, gait analysis is a global term used to encompass several examinations and techniques which study individual components of human gait. Specifically, they include motion capture systems, electromyography, accelerometers, force plates and the use of inertial sensors. In our study, we aimed to analyze kinematic data using the validated Gait Profile Score (GPS), which required an independent capture system to record 3D kinematic data. The most commonly used technologies used for such analysis are motion capture cameras, optoelectronic systems, inertial sensors and electrogoniometers [11]. Our method of choice was an optoelectronic system utilizing passive markers illuminated by infrared light, which were precisely placed above anatomically defined landmarks. We preferred this method to the use of inertial systems such as accelerometers and gyroscopes, which are less comfortable for the patient, more challenging to place above rotatory axes and have slower reaction times compared optoelectronic systems [12]. We also opted for preforming gait analysis in a specialized laboratory rather than a treadmill, as it has been described to support a more natural gait pattern [13].

A fundamental problem of gait analysis is its high sensitivity in detecting abnormal gait patterns caused by various musculoskeletal, neurological or psychiatric pathologies. This makes patient selection a crucial methodological component. As LSS is most commonly diagnosed in the elderly population, a large number of patients had concurrent musculoskeletal diseases and were not eligible for the study. Furthermore, patients with LSS are a heterogenous group with varying degrees of spinal canal stenosis, radicular pain and paresis. As such, we only selected patients with severe spinal canal stenosis (Schizas C or D) and did not include patients with radicular pain or paresis, as these symptoms could by themselves, significantly affect gait patterns [14]. Finally, we excluded patients with LSS above the L3 vertebra, as these cases tend to affect more proximal motion segments of the lower limbs, thus altering their gait stereotype.

### 4.2. Interpretation of Kinematic Gait Parameters

Kinematic gait parameters depict angular changes between two sets of axes, typically in a joint. As angles and motion vectors within a joint dynamically change throughout the gait cycle, description and interpretation of kinematic gait parameters is a challenge. Due to this, several scoring and indexing systems, such as the Gillette Gait Index or Gait Deviation Index have been proposed to aid with their interpretation [15,16]. In our study, we opted for the Gait Profile Score (GPS), a single value describing gait deviation, which has been validated for multiple conditions such as stroke, Parkinson’s disease or multiple sclerosis [17,18,19]. Its main advantage is its composition of nine distinct kinematic variables (GVS), which can be analyzed separately from the global GPS. This gives us insight into specific pathological mechanisms occurring at the level of individual joints. In our study, the overall GPS of patients with spinal stenosis significantly differed from healthy controls, demonstrating major differences in gait patterns between these two groups. In order to identify which motion segments were most affected, we analyzed kinematic graphs of all nine GVS parameters, concluding that three anatomical areas were most affected in patients with spinal stenosis.

The first area with abnormal kinematic parameters was the pelvic region, which demonstrated a lower range of both cranial/caudal tilt and internal/external rotation (Figure 7). This suggests increased pelvic rigidity in the patient group, which is a finding consistent with Bumann et al., who reached similar results in their cohort of 29 patients with LSS [20]. Pelvic rotations are coupled with rotations of the lumbar spine and occur simultaneously as the so-called lumbopelvic rhythm [21]. Posterior pelvic tilt is coupled with lumbar spine flexion, which increases the cross-sectional area of the spinal canal by up to 11% [22]. This is an essential compensation mechanism of patients with lumbar stenosis as it typically provides partial relief of their symptoms. The overall increase in pelvic rigidity may be the result of patients constantly attempting to stabilize their lumbopelvic region to increase spinal canal diameter.

The second area with abnormal kinematic findings was the hip region, which showed decreased hip extension, adduction and abduction (Figure 5 and Figure 6). Decreased hip extension is a finding consistent with Yokogawa et al., who analyzed differences in hip motion between patients with hip osteoarthritis, patients with LSS and healthy controls [23]. Our interpretation is that decreased hip extension is a compensation mechanism caused by anterior shift of the center of gravity due to increased lumbar flexion, which is coupled with semiflexion of the hip joint [24]. This prevents maximal extension in the hip joint, which could result in forward toppling. Patients also demonstrated decreased hip abduction, and to a lesser extent hip adduction, throughout the gait cycle. Hip abductor weakness is a common finding in patients with LSS, as they are innervated predominantly by inferior cauda equina nerve roots [25]. Kim et al. studied coronal plane gait patterns of patients with LSS by performing surface EMG of hip muscles [26]. They found that these patients required increased activation of hip abductors and recruited neighboring quadriceps muscle fibers when performing hip abduction. They concluded that patients with LSS have limited control of each hip abductor, thus recruit more muscles during this activity in an effort to maintain coronal balance. The fact that hip abduction is more strenuous for patients with LSS could explain why they avoid this movement.

The final area of abnormal kinematic findings was the ankle region, where a major decrease in plantar flexion was observed (Figure 8). This decrease is most likely caused by a shortened step length, which does not allow maximal plantar flexion to occur [24]. This is a phenomenon we have described in our article concerning gait analysis of patients with sacroiliac joint dysfunction [27].

### 4.3. Interpretation of Spatiotemporal Gait Parameters

Spatiotemporal gait parameters represent multiple measurements throughout the gait cycle and their sequencing. As their measurement and interpretation is simpler compared to kinematic parameters, changes of these parameters in patients with LSS have been published by multiple studies [3]. Our analysis revealed statistically significant differences of all major spatiotemporal gait parameters between patients and controls. Parameters, which we categorized into the following 3 groups: gait cycle associated (swing phase, stance phase, initial double limb support), distance associated (step length, stride length, step width) and time associated (step times, cadence) parameters.

Considering gait cycle associated parameters, patients with LSS demonstrated a shorter swing phase, longer stance phase and longer initial double limb support. In a gait cycle, the swing phase represents anterior balance loss as weight is transferred from one lower limb to the other. This balance loss must then be compensated by a stable stance phase [24]. As patients with LSS have an anteriorly displaced center of gravity due to lumbar flexion, they require a longer stance phase, longer initial double limb support and a shorter swing phase as compensation mechanisms. Similar findings were demonstrated by Loske et al., who performed gait analysis utilizing inertial sensors [28].

Considering distance associated parameters, patients with LSS were found to have significantly shorter steps and consequently strides compared to healthy controls. This is a consistent finding amongst patients with LSS, which has been described by Sun et al. using motion system analysis and Fujita et al. using the two-step test [29,30]. The authors explain the presence of shorter steps as a result of shorter swing phases due to gait instability [14]. Apart from shorter steps and strides, patients with LSS had significantly greater step width compared to healthy controls. This is a characteristic which has been described by Kim et al. as a compensation mechanism countering excessive trunk sway, a phenomenon found in patients with LSS according to Suda et al. [31]. As our analysis was focused on lower limb parameters, we did not specifically measure trunk sway in our study.

Finally, patients with LSS were found to have significantly altered time associated gait characteristics. Specifically, they had longer step times and a decrease in cadence compared to healthy controls, which resulted in slower gait speed. These results have previously been published by multiple studies such as Peering et al. or Conrad et al., who suggested that a decrease in gait speed and cadence is caused by the presence of lower limb pain and dysesthesias [32,33]. Furthermore, Conrad et al. also found correlations between gait velocity and ODI, showing that this parameter is essential in determining life quality of patients.

### 4.4. Study Limitations

Our study was performed with several limitations. A major limitation is the low number of patients included; however, this was the result of strict patient selection, which is crucial for valid results of gait analysis. Secondly, patients demonstrated various claudication intervals (10–500 m), which meant that their clinical symptoms emerged at various times during the data acquisition. Although all patients underwent a total of 10 walks across the laboratory, patients with shorter claudication intervals could be limited during earlier walking cycles. However, this can be countered using the results of Suda et al., who demonstrate that in patients with lumbar spine stenosis, gait style is altered from the moment they verticalize, prior to the emergence of claudication symptoms [31].

## 5. Conclusions

In conclusion, our study was, to the best of our knowledge, the first to perform objective kinematic gait analysis of patients with LSS via the GPS and compare to matched healthy controls. By utilizing kinematic graphs of individual motion segments, we were able to portray specific motion changes in individual joints and motion planes. This has given us valuable insight into gait pathophysiology of this patient group. Furthermore, spatiotemporal gait parameters were also analyzed, which contributes to the global databank of patient data for this diagnostic entity. In the future, analysis of force moments within individual joints could give insight into secondary degenerative changes occurring in these areas. Additionally, long-term follow-up of postsurgical patients and their gait parameters would allow us to objectify the effect that decompressive surgery has on the gait patterns of patients with LSS.

## Figures and Tables

**Figure 1 sensors-22-01633-f001:**
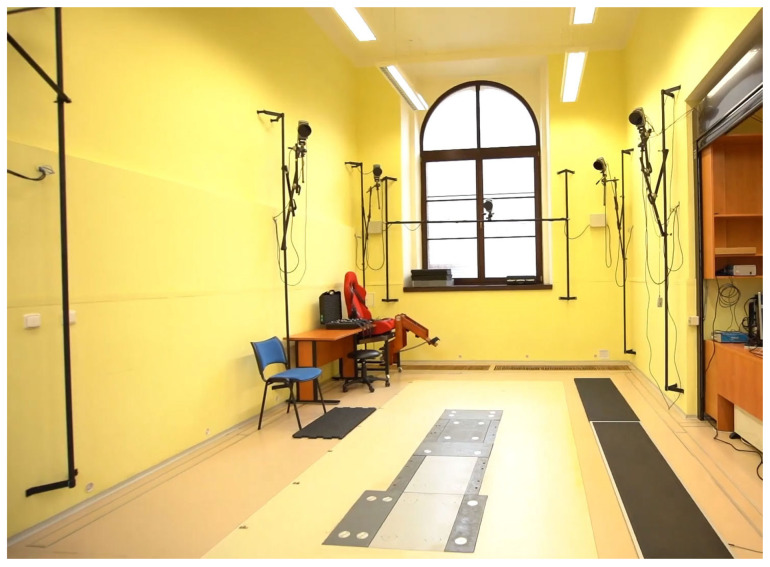
The specialized laboratory with 5 of the 11 infrared cameras Oqus 300 and 300+ visible, placed along the laboratory walls and 2 force platforms (light grey rectangles) placed on the laboratory floor.

**Figure 2 sensors-22-01633-f002:**
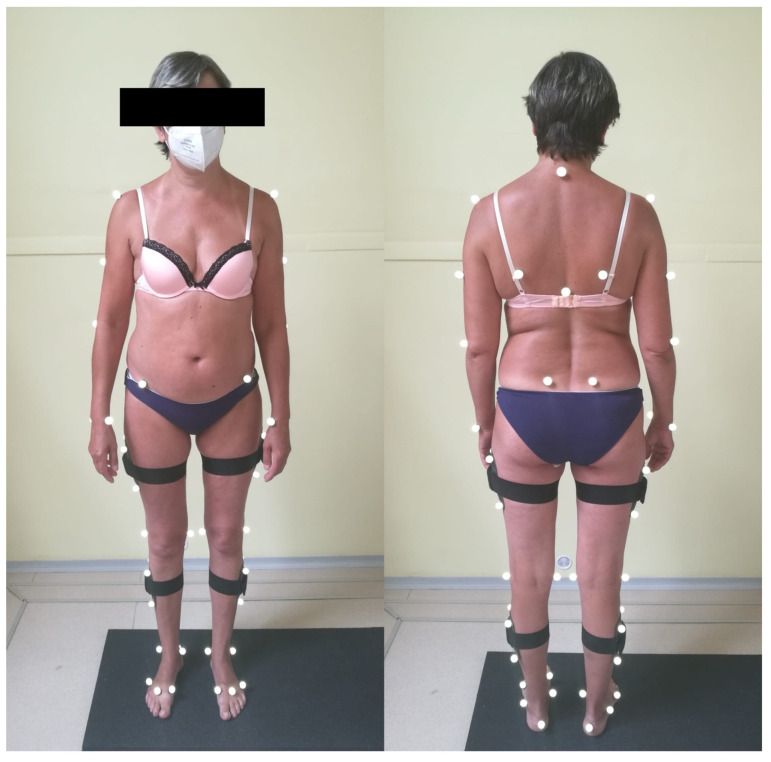
A patient marked with 50 passive reflective markers, placed individually or in clusters of four above major anatomical structures via adhesive tape. Placement was based on identifying anatomical landmarks via the atlas of Sint et al. [7].

**Figure 3 sensors-22-01633-f003:**
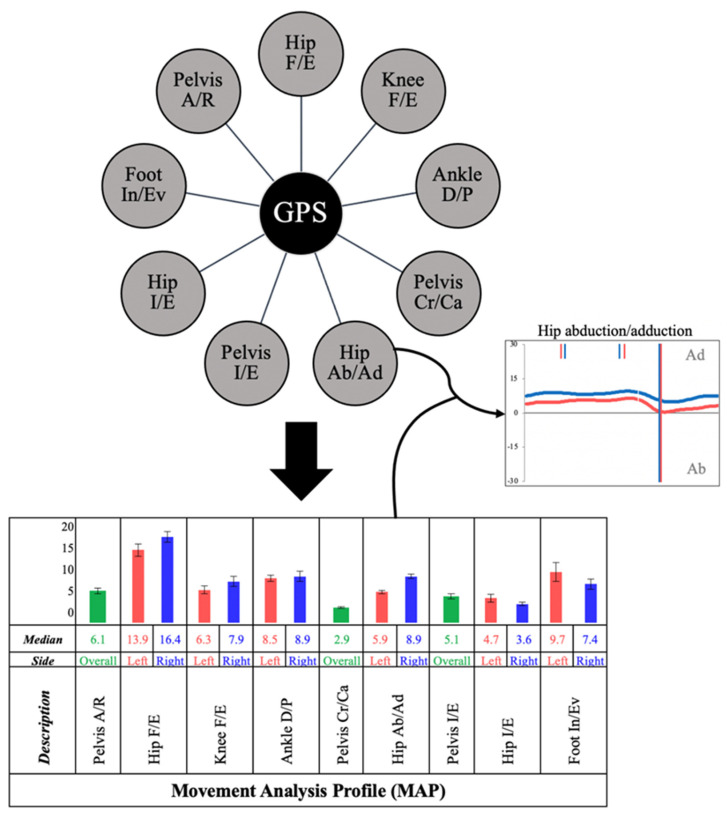
Diagram of the Gait Profile Score (GPS), which is formed of nine Gait Variable Scores (GVS), which generate a Movement Analysis Profile (MAP) of an individual patient. Individual GVS parameters can then be analyzed in detail using motion graphs (for example hip abduction/adduction as shown above).

**Figure 4 sensors-22-01633-f004:**
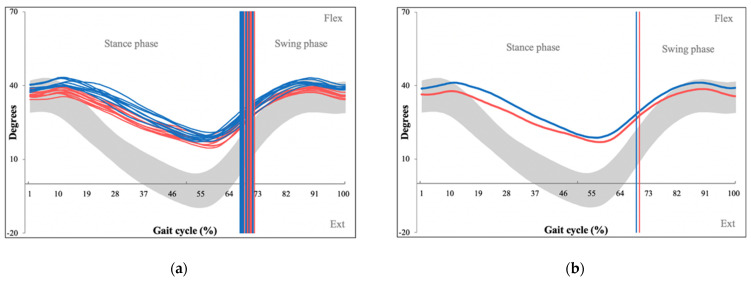
Kinematic consistency motion graphs of hip flexion/extension during 15 gait cycles (**a**) used to generate an average kinematic motion graph (**b**), which then generates the patient’s Gait Variable Score (GVS) for hip flexion/extension. Degrees of movement are shown on the vertical axis whereas the horizontal axis portrays the gait cycle. (Red line—left leg values, blue line—right leg values, grey space—normative values for hip flexion/extension).

**Figure 5 sensors-22-01633-f005:**
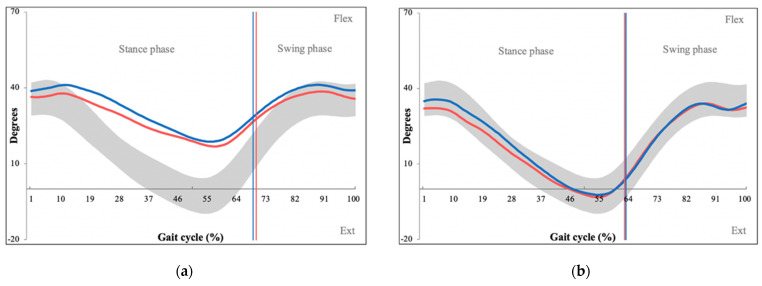
Kinematic graphs of hip flexion/extension of a patient with LSS (**a**) and their matched control (**b**), demonstrating limited hip extension during the stance phase. Degrees of movement are shown on the vertical axis whereas the horizontal axis portrays the gait cycle. (Red line—left leg values, blue line—right leg values, grey space—normative values for hip flexion/extension).

**Figure 6 sensors-22-01633-f006:**
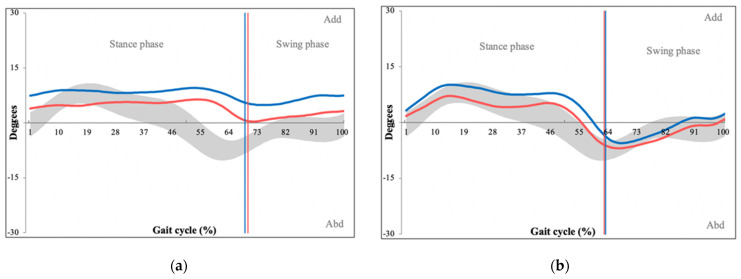
Kinematic graphs of hip abduction/adduction of a patient with LSS (**a**) and their matched control (**b**), demonstrating limited hip abduction throughout the gait cycle. Degrees of movement are shown on the vertical axis whereas the horizontal axis portrays the gait cycle. (Red line—left leg values, blue line—right leg values, grey space—normative values of hip abduction/adduction.)

**Figure 7 sensors-22-01633-f007:**
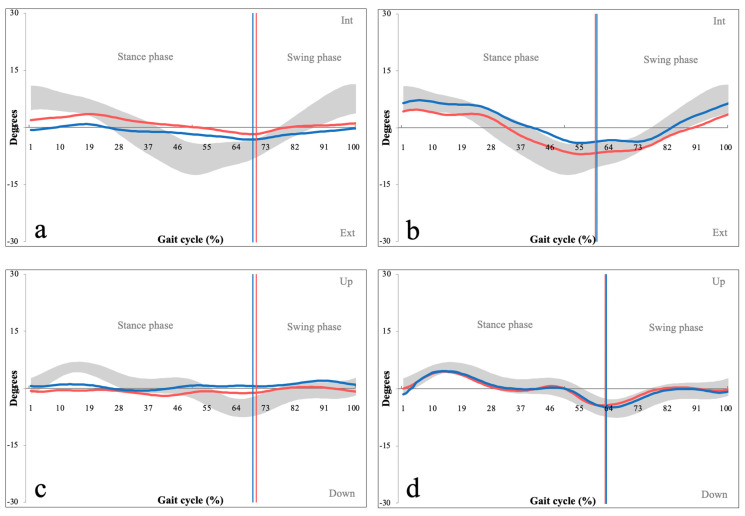
Kinematic graphs of pelvic internal/external rotation of a patient with LSS (**a**) and their matched control (**b**) as well as pelvic cranial/caudal movement of a patient with LSS (**c**) and their matched control (**d**). The graphs demonstrate increased pelvic rigidity throughout the gait cycle. Degrees of movement are shown on the vertical axis whereas the horizontal axis portrays the gait cycle. (Red line—left pelvic values, blue line—right pelvic leg values, grey space—normative values of pelvic internal/external rotation (**a**) and cranial/caudal movement (**b**).)

**Figure 8 sensors-22-01633-f008:**
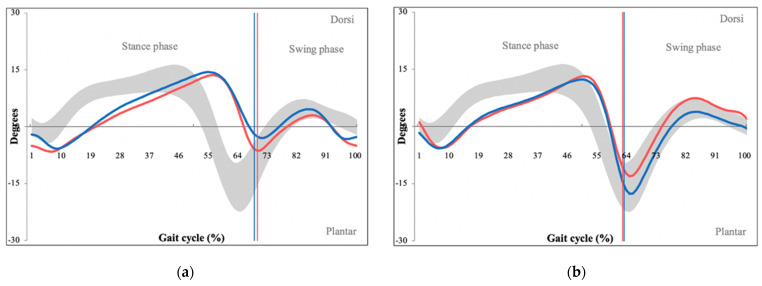
Kinematic graphs of ankle plantar and dorsal flexion of a patient with LSS (**a**) and their matched control (**b**), demonstrating limited ankle plantar and dorsal flexion. Degrees of movement are shown on the vertical axis whereas the horizontal axis portrays gait cycle. (Red line—left leg values, blue line—right leg values, grey space—normative values of ankle plantar/dorsal flexion.)

**Figure 9 sensors-22-01633-f009:**
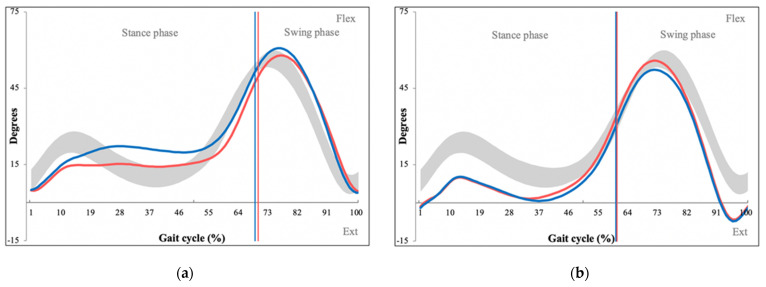
Kinematic graphs of knee flexion and extension of a patient with LSS (**a**) and their matched control (**b**), demonstrating comparable degrees of motion within the knee. Degrees of movement are shown on the vertical axis whereas the horizontal axis portrays gait cycle. (Red line—left leg values, blue line—right leg values, grey space—normative values of knee flexion/extension.)

**Table 1 sensors-22-01633-t001:** Positioning of reflective markers above anatomical structures.

Anatomical Position	Marker Count	Bilateral Marker	Marker Discarded for Gait	Anatomical Position	Marker Count	Bilateral Marker	Marker Discarded for Gait
Trunk				Lower limb			
C7 spinous process	1	No	No	Anterior + posterior superior iliac spine	2	Yes	No
T10 spinous process	1	No	No	Midportion of the lateral thigh	4 (thigh cluster)	Yes	No
Upper limb				Medial + lateral femoral epicondyle	2	Yes	Yes
Inferior angle of the scapula	1	Yes	No	Proximal portion of the lateral shank	4 (shank cluster)	Yes	No
Acromial edge of the scapula	1	Yes	No	Prominence of the medial + lateral malleolus	2	Yes	Yes
Middle of the humerus body	1	Yes	No	Achilles’ tendon insertion	1	Yes	No
Lateral humeral epicondyle	1	Yes	No	Dorsal margin of 1st and 5th metatarsal head	2	Yes	No
Ulnar + radial styloid process	2	Yes	No	Dorsal aspect of 2nd metatarsal head	1	Yes	No

**Table 2 sensors-22-01633-t002:** Spatiotemporal gait parameters of both limbs.

Parameters	Definition	Controls Mean (SD)	Patients Mean (SD	*t*-Test/Welch’s Test (p)
Stride Length (m)	Distance of the full gait cycle from the heel strike of the reference foot to its successive heel strike	1.35 (0.08)	1.00 (0.22)	<0.001
Stride width (m)	Lateral distance between heel centers of two consecutive foot contacts	0.10 (0.02)	0.14 (0.04)	0.004
Step Length side-adjusted (m)	Distance between two successive heel strikes of different feet	0.68 (0.04)	0.50 (0.11)	<0.001
Cadence (steps/min)	The number of steps performed per unit of time	111.2 (3.86)	98.2 (12.51)	0.003
Step time (s)	Time of two successive heel strikes of different feet	0.54 (0.02)	0.62 (0.09)	0.004
Stance time side-adjusted (s)	Time during which the reference foot is in contact with the ground (from heel strike to toe-off)	0.66 (0.03)	0.85 (0.17)	<0.001
Swing time side-adjusted (s)	Time in which the reference foot is not contact with the ground (from toe-off to heel strike)	0.42 (0.02)	0.40 (0.04)	0.04
Initial double-limb support (s)	Time during which both feet are in contact with the ground in the initial phase of the gait cycle	0.12 (0.01)	0.22 (0.08)	<0.001

**Table 3 sensors-22-01633-t003:** Kinematic angular gait parameters of both limbs.

Parameter	Definition	Controls Mean (SD)	Patients Mean (SD)	*t*-Test/Welch’s Test (p)
**Gait Profile Score (GPS)**	A summarized validated index used to compare a patient’s gait pattern with normative data	**6.44 (1.00)**	**8.65 (2.44)**	**0.004**
Pelvic anteversion/retroversion	Describes pelvic tilting around the femur heads in the sagittal plane	4.24 (2.55)	5.41 (3.16)	0.273
Pelvic cranial/caudal tilt	Describes cranial/caudal pelvic tilt, measured against a horizonal plotline the frontal plane	**1.82 (0.54)**	**3.20 (2.24)**	**0.034**
Pelvic internal/external rotation	Describes rotational pelvic motion around the lumbar column axis in the axial plane	**3.44 (0.87)**	**5.31 (1.41)**	**<0.001**
Hip flexion/extension	Describes flexion/extension of the hip joint in the sagittal plane	**6.22 (2.79)**	**11.31 (6.50)**	**0.012**
Hip abduction/adduction	Describes medial/lateral hip movement in the frontal plane	**3.73 (1.19)**	**5.73 (2.57)**	**0.012**
Hip internal/external rotation	Describes rotational hip movement around the femur head in the axial plane	7.58 (4.17)	9.49 (4.95)	0.264
Knee flexion/extension	Describes flexion/extension of the knee joint in the sagittal plane	9.74 (2.66)	8.46 (2.58)	0.192
Ankle dorsal/plantar flexion	Describes flexion/extension of the crural joint in the sagittal plane	**4.80 (1.32)**	**8.28 (2.45)**	**<0.001**
Foot inversion/eversion	Describes medial/lateral movement of the foot in the frontal plane	6.49 (1.89)	7.20 (2.52)	0.394

**Table 4 sensors-22-01633-t004:** Patient characteristics (*n* = 15).

**Mean Patient Age (SD)**	**62.3 (10.6)**	**Stenotic Level**	
**Gender (male:female)**	11:4	L3/4	1
**Mean Body Mass Index (SD)**	32.0 (5.3)	L4/5	5
**Mean Oswestry Disability Index (SD)**	52.4 (12.1)	L3–5	7
**Radicular hypesthesia**	7 (46.7%)	L4-S1	2
**Claudication interval**		**Schizas degree of stenosis**	
Less than 50 m	7	C	3
50–200 m	6	D	12
200–500 m	2		

## Data Availability

The data presented in this study are available on request from the corresponding author. The data are not publicly available as they contain personal and medical data of the analyzed subjects, which our ethics committee had not allow to release.

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
