# Peer review of "Quantitative Gait Analysis of Patients with Severe Symptomatic Spinal Stenosis Utilizing the Gait Profile Score: An Observational Clinical Study"

_sensors, 2022, doi:10.3390/s22041633_

Round 1

Reviewer 1 Report

The authors report a gait analysis of patients with Lumbar Spinal Stenosis (LSS) in comparison to healthy controls using both spatiotemporal gait characteristics as well as the Gait Profile Score and kinematic parameters. I have a few minor concerns with the methods reporting (see below) but otherwise the investigation seems to be sound and the conclusions are supported from the results. My larger concern, which I leave to the discretion of the Editor, is whether this research report is appropriate for this special issue of Sensors. Per the special issue information, the issue's aim is to address topics related specifically to the innovation or use of sensors for investigating human biomechanics (e.g., "how to warrant correct positioning..", "how smart does a sensor need to be..", etc.). This paper describes the results of an experiment that made standard use of commercially available motion capture technology to perform gait analyses on a clinical population. This method, from a sensors technology perspective, is not novel. The novel contribution of the paper lies in the observations made about Lumbar Spinal Stenosis, which is not trivial but perhaps not appropriate for the special edition to which it was submitted.

Minor concerns with the method:

How is this a prospective study? A prospective study is a very specific type of study and the authors make no mention of how their experiment is prospective in nature. There is brief mention in the abstract about quantifying surgical outcomes in the future, and the authors mention that all patients underwent surgical decompression after the experiment. But nothing is mentioned about how the current study is prospective.

The hypotheses come across a bit vague. Did the authors have any a prior expectations as to how the gait patterns of patients with severe LSS would differ from healthy controls (i.e., any directional hypotheses)? Or was this merely an exploratory study?

For reproducibility, it is not sufficient to say that markers were placed "by an experienced anatomist" on "major anatomical structures". Detail should be provided about the exact anatomical placement of markers (e.g., lateral acromion, lateral epicondyle, radial styloid), which is pertinent for considering the joint angle measures. Given the large number of markers, it may be necessary to put this in a table, or a figure caption, but it can typically be summarized in a sentence or two. If a well-cited marker set was used as a reference  (e.g., Cleveland Clinic), this should be mentioned as well.

More information is also needed about the experimental task/design, and data processing. First, the data were sampled at 100 Hz. Sometimes based on camera resolution there is noise in recorded marker position that needs to be filtered. Given the low frequency range of gait cycles, a low-pass filter (cut off 4-6 Hz) would be fine to use to reduce noisiness. Was any filtering performed? Second, the patient was "asked to walk at a natural pace across the length of the laboratory" ... "a total of 10" times. This resulted in roughly how many gait cycles per patient? Are there descriptive data on how the natural pace differed within and between subjects? It sounds like the authors averaged gait cycles not only across a single trial, but also across the 10 repeated trials. It might be good to verify that gait patterns were consistent across the 10 trial repetitions by indicating that averaged gait cycles for each trial are not substantially different.

Reviewer 2 Report

The proposed study was designed by the authors to evaluate the spatiotemporal and kinematic parameters extracted from patients with lumbar spine stenosis using the gait profile score. The manuscript is well written and the references cited are up to date. However, the structure of the manuscript and the introduction/discussion sections should be improved. As stated by the Authors in the Introduction, several previously published studies have evaluated the gait performance of patients with LSS through spatiotemporal parameters (e.g. stride length, cadence, stance/swing duration, etc.) and also through kinematic data (eg Ref. [4] and [5]). Therefore, I have a major concern about the novelty and originality of this work. In my opinion, a quantitative 3D motion analysis of the effects of decompression surgery on gait performance would have been much more original and interesting for readers. 

Round 2

Reviewer 1 Report

The authors addressed my prior comments. I find the quality of the manuscript to be improved and but have a few final comments, aimed at improving the manuscript quality, that should be addressed before acceptance.

1. I appreciate the added detail to the opening paragraph. It would be good to add some additional clarification to a few more sections in the paper. First, in the intro paragraph, on 1) the distinction between spatiotemporal gait parameters and kinematic parameters, as this distinction is not necessarily intuitive, and 2) an example of how spatiotemporal gait parameters do not provide information about individual motion segments and how this is crucial for LSS assessments. This could be accomplished in just one or two sentences, giving an example of a spatiotemporal parameter and the information it provides, then explaining how it is not sufficient for some clinical question, and then citing a relevant example kinematic parameter that can satisfy that question.

To the same end, the authors state at the end of the introduction that the novel contribution of the paper is the use of a gait scoring system where prior studies have not. Adding a sentence or two clarifying what is important about this contribution (e.g., what information does it provide that was lacking before) will help the reader understand the novelty and value of the work.

In the aims and hypothesis section, given that this is an exploratory study, it would be helpful to briefly introduce how the validated gait scoring system can quantify the differences as stated in lines 58-59. The authors do detail this in section 2.4 but it will still be nice to prime the reader for this info with a brief summary statement in section 2.1.

Finally, in the Discussion, it is good practice to start first and foremost with a sentence or two (i.e., below 4. Discussion but prior to 4.1 Gait Analysis Technique and Patient, if possible) re-stating the purpose of the paper and summarizing main takeaways of the results. This will help remind and refocus the reader to the pertinent discussion points.  

2. I appreciate the additional tables and the additional figures. One small question about Figs. 4—9: what do the vertical red/blue lines located at the ~60-70% gait cycle mark represent? They seem to represent data given than they have different values.

3. Another round of proofreading is necessary; I caught a few grammar typos here and there, especially in the added text.
